# Selected Psychosocial Factors, Nutritional Behavior, and the Analysis of Concentrations of Selected Vitamins in Patients with Cardiovascular Diseases

**DOI:** 10.3390/nu16121866

**Published:** 2024-06-14

**Authors:** Anna Krystyna Główka, Magdalena Kowalówka, Paweł Burchardt, Anna Komosa, Łukasz Kruszyna, Mirosław Andrusiewicz, Juliusz Przysławski, Marta Karaźniewicz-Łada

**Affiliations:** 1Department of Bromatology, Poznan University of Medical Sciences, 60-806 Poznań, Poland; mkowalowka@ump.edu.pl (M.K.); jprzysla@ump.edu.pl (J.P.); 2Department of Hypertensiology, Angiology and Internal Medicine, Poznan University of Medical Sciences, 61-848 Poznań, Poland; pburchardt@ump.edu.pl (P.B.); komosa.ania@ump.edu.pl (A.K.); 3Department of Cardiology, Józef Struś Hospital, 61-285 Poznań, Poland; 4Department of Vascular and Endovascular Surgery, Angiology and Phlebology, Poznan University of Medical Sciences, 61-848 Poznań, Poland; lukaszkruszyna@ump.edu.pl; 5Department of Cell Biology, Poznan University of Medical Sciences, 60-806 Poznań, Poland; andrus@ump.edu.pl; 6Department of Physical Pharmacy and Pharmacokinetics, Poznan University of Medical Sciences, 60-806 Poznań, Poland; mkaraz@ump.edu.pl

**Keywords:** psychosocial factors, cardiovascular diseases, eating behavior, plasma levels of vitamins, fat-soluble vitamins

## Abstract

Cardiovascular diseases (CVD) are the leading cause of death worldwide, influenced by the interaction of factors, including age, sex, genetic conditions, overweight/obesity, hypertension, an abnormal lipid profile, vitamin deficiencies, diabetes, and psychological factors. This study aimed to assess the relationships between psychosocial and nutritional factors in a group of 61 patients with CVD (i.e., atherosclerosis, hypertension, ischemic heart disease, and myocardial infarction) and their possible impact on the course of the disease. The plasma concentrations of vitamins A, E, D, and β-carotene were determined using validated HPLC-MS/MS, while the lipid profile was analyzed enzymatically. Psychosocial factors and nutritional behaviors were assessed using author-designed questionnaires. Over 50% of patients had 25-OH-D3 and retinol deficiencies, while >85% of patients exhibited significant deficiencies in α-tocopherol and β-carotene. The lipid profile showed no specific relationship with any particular CVD. Dietary behavior minimally impacted biochemical parameters except for higher β-carotene concentrations in the group with higher fruit and vegetable intake. The negative impact of the CVD on selected parameters of quality of life was noticed. To increase the effectiveness of the prevention and treatment of CVD, the need for interdisciplinary cooperation observed between doctors, psychologists, and specialists in human nutrition seems to be justified.

## 1. Introduction

The development of cardiovascular disease (CVD) is strongly related to several risk factors, including an unhealthy diet, a lack of physical activity, smoking, being overweight, and diabetes. The contribution of psychosocial risk factors to the pathogenesis of CVD should be considered in individual patients, and appropriate interventions should be proposed [1]. The psychosocial factors presented in Figure 1 are associated with an approximately 1.5-fold increase in the risk of cardiovascular events. Several mechanisms (biological and behavioral pathways) link these risk factors with poor prognosis and contribute to adverse clinical outcomes in people with CVD [2].

One such factor is the patient’s personality. Among the four conceptualizations of personality traits, namely, A, B, C, and D, the last one proved to have the highest impact on CVD development. Type D personality is defined as the co-occurrence of negative affectivity and social inhibition, significantly increasing the risk of CVD and showing a higher CVD mortality rate compared to the other types. In addition, personality type D‘s characteristics negatively impact the rehabilitation of patients with CVD due to the high severity of somatic symptoms and symptoms of exhaustion [3]. These patients also experience more anxiety and depression and are in poorer mental health, which, in turn, translates into a poorer quality of life [4,5].

Depression contributes to the severity of atherosclerosis, which increases the risk of myocardial infarction and death from cardiovascular causes. In the case of people with coronary heart disease (CHD), type D personality combined with age > 55 years, reduced left ventricular function, poor exercise tolerance, and depression lead to a lack of the expected therapeutic response [6].

An additional factor related to the incidence of CVD and the associated high mortality rate is the current progress of civilization, which has resulted in lifestyle changes and increased susceptibility to stressors, as well as low socioeconomic status, a lack of social support, and withdrawal from life [7].

There is evidence that anxiety is a risk factor for hypertension, excessive obesity, and smoking, which can accelerate atherosclerosis. Anxiety and chronic stress lead to activation of the hypothalamic–pituitary–adrenal axis and increased levels of catecholamines and cortisol, resulting in dysregulation of the autonomic nervous system and a cascade of further effects that may increase the risk of death from CVD, including coronary heart disease, stroke, and heart failure [8,9]. Approximately 30% of all CVD deaths in western countries are due to nicotine addiction. The risk of death from smoking increases with the simultaneous presence of hypertension, hypercholesterolemia, and peripheral vascular disease [10].

Low physical activity and the associated overweight or obesity are among the most common factors increasing CVD risk. They also cause deterioration of physical and mental health, which also translates into a worse perception of selected aspects of quality of life by people affected by them [11].

Lifestyle diseases, such as diabetes, hypertension, and hyperlipidemia, adversely affect the heart and cardiovascular system. These diseases are also primarily caused by an unhealthy diet characterized by a high content of saturated fatty acids, cholesterol, simple carbohydrates, protein, and excessive sodium intake. However, the diet lacks unsaturated fatty acids, dietary fiber, complex carbohydrates, and some vitamins and minerals, resulting from the low consumption of whole grain products, fruits and vegetables, legumes, fish, and nuts [12,13].

Nutrients that play an important role in the pathogenesis of CVD include vitamins A, D, E, and β-carotene. A factor that can significantly influence the inhibition of atherosclerotic lesions is the vitamin D status in the body [14,15]. It has been proven that the relationship between body weight and vitamin D intake is inversely proportional, and low vitamin D concentration is an independent risk factor for obesity, which, in turn, promotes the occurrence of CHD [16,17]. Its deficiency is significantly associated with the risk of congestive heart failure, heart attack, and ischemic heart disease, while retinol, α-tocopherol, and β-carotene play an important role in inhibiting the formation of atherosclerotic plaque [18,19,20].

The aim of the study was to assess the relationship between psychological and dietary factors in the group of patients with CVD (atherosclerosis, hypertension, ischemic heart disease, and myocardial infarction) and their possible impact on the course of the disease.

## 2. Materials and Methods

### 2.1. Study Group

The study group included 192 patients treated in the Department of Cardiac Intensive Care and Internal Medicine, Heliodor Święcicki Clinical Hospital of Poznan University of Medical Sciences, and the 1st Department of Cardiology and Department of General and Vascular Surgery, both at the Transfiguration Clinical Hospital of Poznan University of Medical Sciences. The criterion for including a patient in the study was a previously diagnosed cardiovascular disease: atherosclerosis, arterial hypertension, ischemic heart disease, and infarcts of various organs. The exclusion criteria included a number of platelets below 100,000/mm^3^, a serum creatinine concentration above 2 mg/dL, liver impairment, acute coronary syndrome in the last three months before admission to the hospital, and active cancer. A group of 105 patients met the above criteria. In this group, in the first stage, the lipid profile, namely, low-density lipoprotein (LDL), high-density lipoprotein (HDL), total cholesterol (TC), and triglycerides (TG), were assessed, and plasma concentrations of retinol, α-tocopherol, β-carotene, 25-hydroxyvitamin D2 (25-OH-D2), and 25-hydroxyvitamin D3 (25-OH-D3) were determined. The biological material for the study consisted of venous blood (5 mL) collected from the patients using EDTA aspiration-vacuum sets.

The next stage of the study was the assessment of selected psychosocial criteria and nutritional behaviors (discussed in more detail in Section 2.2.1 and Section 2.2.2), which, according to current knowledge, may relate to the course of treatment of patients with cardiovascular disease. The research tool was an original questionnaire containing questions from the area covered by the research. Sixty-one patients declared further participation in the study and filled out the questionnaire. The study was conducted according to the guidelines of the Declaration of Helsinki and approved by the Institutional Review Board of Poznan University of Medical Sciences (protocol code nos.: 273/15 and 644/15). A diagram of the study is shown in Figure 2.

### 2.2. Research Methodology

#### 2.2.1. Assessment of Selected Psychosocial Factors and Lifestyle Parameters of Patients with Cardiovascular Diseases

An original survey questionnaire was used to assess selected psychosocial factors, lifestyle parameters, and the socioeconomic situation of patients with cardiovascular diseases. Questions regarded, among others, place of residence, education, age, stimulants used (e.g., drugs and alcohol), physical activity, pain sensation, and social contacts.

#### 2.2.2. Assessment of Eating Behavior in Patients with Cardiovascular Diseases

An original survey questionnaire was used to assess eating behavior. Questions regarded, among others, food preferences, frequency of consumption of selected food products, type of food products consumed, and the use of a properly balanced diet.

#### 2.2.3. Anthropometric Indicators

Weight and height indices (body height, body weight, and BMI value) were used to assess the nutritional status of the studied group of patients.

#### 2.2.4. Determination of the Lipid Profile

The biochemical tests were performed in laboratories at the clinical units. They included the determination of TC, HDL, LDL, and TG, in accordance with the methodology applicable in a given laboratory [21]. Briefly, the enzymatic-colorimetric method was used to determine TC, HDL, and TG using a Cobas c 501 analyzer (Roche Diagnostics, Poznań, Poland). To determine the concentration of cholesterol in the LDL fraction, Friedewald’s formula was used [22]: “LDL = TC − (HDL + TG/5)”. The formula can only be used when the TG concentration is below 4.5 mmol/L (400 mg%). If the TG concentration exceeded this value, the LDL cholesterol concentration was measured directly. The concentration of non-HDL cholesterol was calculated using the formula “Non-HDL cholesterol [mg/dL] = TC − HDL”.

#### 2.2.5. Determination of the Concentration of Selected Vitamins in the Plasma of Patients with CVD Using the Validated UPLC-MS/MS Method

The determination of the vitamin concentrations in patients’ plasma was carried out using the previously developed and validated UPLC-MS/MS method [23]. The analytes, including retinol, α-tocopherol, 25-OH-D2, 25-OH-D3, and β-carotene and internal standards α-tocopherol-d6, 25-OH-D3-d6, and retinol-d5, were analyzed on the Nexera UPLC liquid chromatograph (Shimadzu, Kyoto, Japan) equipped with a thermostated autosampler (SIL-30AC) and a degasser (DGU-20A5) and coupled with a triple quadrupole mass spectrometer LCMS-8030 (Shimadzu, Kyoto, Japan). The chromatographic separation was carried out in a Kinetex F5 analytical column (10 mm × 2.1 mm, 2.6 µm) connected to a pre-column (Phenomenex, Torrance, CA, USA) using the mobile phase, which was a mixture of water (A) and methanol (B), containing formic acid at a concentration of 0.1% (*v*/*v*). The MS/MS detection was performed using electrospray ionization in positive ion mode (ESI+) at *m*/*z* transitions characteristic for the compounds. The linearity of the UPLC-MS/MS method was confirmed at the concentration levels as follows: 0.02–2 μg/mL for retinol, 2–100 ng/mL for 25-OH-D2 and 25-OH-D3, 0.5–20 μg/mL for α-tocopherol, and 0.05–3 µg/mL for β-carotene. The recovery of analytes from plasma was as follows: for α-tocopherol 86.8–91.8%, for retinol 77.4–87.9%, for 25-OH-D2 67.5–69.4%, for 25-OH-D3 68.3–73.1%, and for β-carotene 36.1–45.7%. The inter- and inter-day precision was in the range of 4.56% for 25-OH-D2 to 18.9% for β-carotene, while the accuracy of the method ranged from 1.77% for retinol to 17.4% for β-carotene.

#### 2.2.6. Statistical Analysis

The conformity of the empirical data distributions to the normal distribution was verified by the W Shapiro–Wilk test. Homogeneity of variances was verified using the Fisher–Snedecor test (for two variables) and Levene’s test (for more than two groups). Analysis of differences for variables with a normal distribution was performed using Student’s *t*-test (for two groups) and one-way analysis of variance (ANOVA) with Tukey’s HSD (Tukey’s honest significant difference test for more than two groups) post-hoc test. If the homogeneity of variance assumption was not met, the Student’s *t*-test with independent variance estimation and the *F*-Welch test were used, respectively. The Mann–Whitney *U* test and Kruskal–Wallis test with Dunn’s post-hoc test were used for inconsistencies with parametric distribution and ordinal scale variables.

In the analyses of the relationship between nominal variables and ordinal variables with a small number of measurement points, χ^2^ tests were used, using Cochran’s conditions to select the appropriate correction (Yates, Fisher’s, Cochran–Mantel–Haenszel, NW, and Fisher–Freeman–Halton).

Ward’s method of agglomerating variables and determining clusters based on the similarity of responses with the Euclidean distance determination was used to reduce and classify questionnaire-based patient data. A tree and scree plots were used to determine the number of clusters for each agglomeration. Patients were assigned to individual clusters based on tying common cases into groups. Due to the large amount of data, we decided to carry out the analysis based on the reduced data using agglomerations of objects and features. Agglomerations forming two to three clusters were used, thus allowing observation of the influence of multiple parameters on each other. Differences in group sizes in the various analyses are due to error and/or non-response.

## 3. Results

### 3.1. Characteristics of Patients

Table 1 shows the general anthropometric characteristics of the studied group of patients, sociodemographic parameters, where residence and education were assessed, and the type, duration, and number of diagnosed cardiovascular diseases. Of the study group, 36% were women (*n* = 22) aged 65.0 ± 9.13 years, and 64% were men (*n* = 39) aged 61.1 ± 3.17 years. The average body weight for both men and women was about 83 kg (83.5 ± 13.6 and 83.0 ± 10.0 kg, respectively). The BMI of most subjects was more than 25 kg/m^2^, and, according to the standards, they were indicated as overweight, especially in women (28.9 ± 3.23 kg/m^2^). The difference in BMI between women and men was statistically significant (*p* = 0.036). A significant majority of respondents (more than 70% of both men and women) indicated a city as their place of residence. The remainder lived in villages (*p* = 0.045). Moreover, 44% of the surveyed people had higher education. Taking into account the number of diagnosed diseases (1–4), in the case of women, 50% of the respondents indicated two diseases, while 36% indicated only one disease. A similar situation was observed for men (46%: 2 diseases and 33%: 1 disease). Three/four disease units were indicated by 14% of women and 21% of men (*p* > 0.05). Another parameter that was analyzed was the type of disease (atherosclerosis, hypertension, ischemic heart disease, and infarctions of various organs). In the studied group of patients, three disease entities predominated. These were arterial hypertension, ischemic heart disease, and infarctions of various organs. There was a significant difference in the prevalence of atherosclerosis between men and women. Men were significantly more likely to report the presence of atherosclerosis (*p* = 0.042). Hypertension affected 82% of women and 69% of men (*p* > 0.05), and ischemic heart disease affected 77% of women and 72% of men (*p* > 0.05). The percentage of patients with previous infarctions of various organs was 18% for women and 28% for men. More than 80% of the women and almost 60% of the men surveyed indicated a duration of diagnosed cardiovascular disease of 4 years or more. However, these differences were not statistically significant (*p* > 0.05) (Table 1).

We also compared the data on plasma vitamin concentrations and the analysis of lipid profiles in groups divided according to sex. Differences were observed between men and women in retinol, HDL, and %HDL levels (Table 2).

Figure 3 shows the percentage distribution of plasma concentrations of the tested vitamins in the analyzed group of subjects in relation to reference values. Deficiency of 25-OH-D3 and retinol was found in about 55% of the subjects. In addition, severe deficiency of α-tocopherol (98.4%) and β-carotene (86.9%) was demonstrated. The values of the subjects’ lipid profile parameters were also analyzed (Figure 4). Considering the reference values, it turned out that most patients had normal levels of TC, HDL, and TG. However, low levels of LDL and non-HDL were observed.

### 3.2. The Prevalence of Cardiovascular Diseases

The agglomeration method was used to reduce and classify patient interview data. Two clusters were identified based on comorbidity variables. None of the planned agglomerations was significantly associated with gender (*p* > 0.05). Therefore, individual agglomerations were not further analyzed in groups by gender. The χ^2^ statistic was determined for the clusters to estimate differences in disease prevalence.

Cluster 1 (*n* = 26) included patients with hypertension and ischemic heart disease but without atherosclerosis and infarcts of various organs. Cluster 2 (*n* = 35) included subjects that indicated co-occurrence of atherosclerosis or infarcts of various organs with any of the other diseases. The analysis of the prevalence of cardiovascular diseases among the subjects is shown in Appendix A. Statistically significant differences were shown between the cases analyzed (*p* < 0.016).

Vitamin and lipid profile analyses were done for all disease entities, and a comparison of results due to the presence of atherosclerosis is included as an example. The plasma vitamin concentrations and lipid profiles did not differ between patients divided according to the criteria of the presence of arterial hypertension, ischemic heart disease, and infarcts of various organs.

Table 3 shows the results of plasma concentrations of vitamins and the lipid profiles of patients according to the presence or absence of atherosclerosis. It showed that those with atherosclerosis had significantly higher retinol concentrations (*p* = 0.016). In contrast, plasma concentrations of vitamin D metabolites, α-tocopherol, and β-carotene were lower compared to subjects without the disease. However, there were no statistically significant differences (*p* > 0.05).

The results showed that those with atherosclerosis were not statistically significantly different from patients without the disease in terms of total cholesterol, HDL fraction cholesterol, LDL fraction cholesterol, triglycerides, and calculated %HDL and non-HDL values (*p* > 0.05), although the HDL and non-HDL values in those with atherosclerotic lesions were on the borderline of normal.

Taking together the conjunction of sex and atherosclerosis, arterial hypertension, ischemic heart disease, and infarcts of various organs, we observed differences only in atherosclerosis and ischemic heart disease at the retinol level. In the first case, females without atherosclerosis had lower retinol levels. In the case of ischemic heart disease, men without ischemic heart disease had significantly higher retinol levels than women with and without the disease. However, because of the small number of observations in these groups.

### 3.3. Analysis of Selected Lifestyle Factors of Stimulants and Physical Activity in the Study Group of Patients

Appendix A presents the analysis of physical activity and stimulant use among the subjects. Based on the above variables, two clusters were distinguished. Cluster 1 (*n* = 23) assigned non-smokers and past smokers, while cluster 2 (*n* = 39) assigned occasional smokers and addicts. Patients assigned to the clusters did not differ in alcohol consumption and physical activity (*p* > 0.05), while statistically significant differences were noted between the study groups regarding tobacco product smoking (*p* < 0.001).

Table 4 shows the results of plasma concentrations of selected vitamins and lipid profiles of patients classified into clusters according to lifestyle parameters. Our study showed that the concentration levels of retinol, α-tocopherol, 25-OD-D2, 25-OH-D3, and β-carotene did not depend on the selected lifestyle factors (*p* > 0.05). In the case of lipid profile parameters determined in patients divided into clusters according to selected lifestyle parameters, there were no significant differences between both groups (*p* > 0.05).

#### 3.3.1. The Current Health and Psychophysical Functioning

The analysis of current health status and psychophysical functioning among the subjects is shown in Appendix A. Cluster 1 (*n* = 17) included patients with no significant impact of health status on functioning in daily life. Respondents assigned to cluster 2 (*n* = 44) indicated limitations in acting, such as fast fatigue, worse mood, problems with performing daily activities, and the need to reduce working time. Statistically significant differences between clusters were found for all analyzed parameters (*p* < 0.001; *p* = 0.003).

It was shown that patients classified in cluster 1 had lower concentrations of 25-OH-D3, α-tocopherol, and β-carotene but higher concentrations of retinol compared to patients in cluster 2. However, the differences were not statistically significant for selected vitamins (*p* > 0.05). In addition, patients from both groups differed significantly in lipid profile, which was reflected by higher %HDL, and lower LDL and non-HDL values in patients from cluster 2 (Table 5).

#### 3.3.2. The Analysis of the Subjective Emotional States

The analysis of the subjective emotional states is shown in Appendix A. Two clusters were again distinguished based on the variables of mental feelings related to the illnesses of the respondents. Cluster 1 (*n* = 19) was assigned cases with reduced capacity for comfort and less energy/willingness to live, not only indicating calmness and feelings of happiness but also not showing illness-related breakdown. Cluster 2 (*n* = 41) included patients who showed limitations in functioning. Statistically significant differences were found between clusters for all parameters analyzed, except for increased energy and will to live (*p* = 0.018; *p* < 0.001).

Comparing the clusters determined by the subjects’ emotional states, it was shown that patients classified in cluster 1 have higher plasma concentrations of 25-OH-D3 metabolites and lower concentrations of retinol and α-tocopherol. However, these differences were not statistically significant. In addition, it was shown that subjects included in the clusters analyzed were characterized by mean lipid profile values that fell within reference ranges (Table 6).

However, no significant differences were shown with respect to the biochemical parameters, both vitamins and lipid profile parameters (*p* > 0.05).

Moreover, it has been tested whether a negative emotional state is associated with more severe CVD. The selected disorders did not influence the emotional state as analyzed separately or in clusters.

### 3.4. Characteristics of Selected Eating Behaviors

Food group consumption was determined, assuming 11 groups of products, which were analyzed based on a 5-degree hedonic scale: “I don’t really like; I don’t like; I don’t care; I like, I like it a lot”. Consumption preferences were analyzed based on an agglomeration diagram, from which two main clusters were extracted (Table 7). Cluster 1 (*n* = 26) was characterized by similar percentages of those who preferred (liked and very much liked) meat and meat products, milk and dairy products, and light and dark bread, while the preferences of dislikers and indifferent were for groats, legumes, and dark bread (20% of respondents). Cluster 2 (*n* = 35) was characterized by a similar percentage of those who preferred meat and meat products, milk and dairy products, and a higher consumption of fish, groats, legumes and vegetables, and fruits. Statistically significant differences between clusters were found for all studied parameters except for consuming meat and processed meat products, dairy products, and pasta.

Table 8 shows the results of plasma vitamin concentrations and parameters of lipid protein of the examined people assigned, depending on dietary preferences, to clusters 1 or 2. Significantly higher levels of β-carotene (*p* = 0.034) occurred in the group classified as cluster 2, characterized by a higher consumption of fruit and vegetables. The lipid profile results depending on dietary preferences showed that a statistically significantly higher %HDL level was found in the group classified into cluster 1 in contrast to cluster 2 (37.6 ± 10.6% vs. 31.3 ± 9.60, respectively; *p* = 0.041). No significant differences were found in the remaining parameters (*p* > 0.05).

## 4. Discussion

Cardiovascular diseases are one of the most severe health problems worldwide. Their occurrence is primarily related to lifestyle. The prevention and treatment of cardiovascular diseases, the impact of nutritional behavior and physical activity on the risk of these diseases, and their course are well documented [24,25,26,27,28,29]. However, there is little research on the additional impact of psychosocial factors. This is the first study conducted in the Polish population aimed at determining the relationship between psychosocial factors, diet, plasma concentration of selected vitamins, and lipid profile and their possible impact on the course of CVD [30,31].

### 4.1. Anthropometric Parameters

We showed that women had a significantly higher BMI value than men. In the case of chronic diseases, long-term adiposity status, particularly middle-life weight gain, has a significant adverse effect. Patterns of weight change in mid-to-late adulthood are sex-specific. Women undergo significant weight and body fat distribution changes around menopause [32,33]. The loss of estrogen following menopause eliminates protections against metabolic dysfunction, mainly due to its role in the health and function of adipose tissue. In addition, menopause is associated with reduced physical activity, which could potentially exacerbate the deleterious cardiometabolic risk profile accompanying menopause [34]. Obesity polygenic scores are also more strongly associated with weight-change trajectories in women than men [29].

### 4.2. Socioeconomic Status and Psychosocial Factors

The rural area was inhabited by about 25% of the respondents, and 75% lived in cities. Significantly more men indicated the town as their residence. A study by Kurpas et al. [35] assessing the quality of life of elderly patients showed significantly lower quality of life in patients living in less populated areas.

The group with higher education dominated, although it was less numerous in the case of women. Increased education and income most often correlate with a decreased risk of coronary heart disease [36]. Kurowska et al. [37] showed that the higher the education, the higher the quality of life and the higher the level of health behaviors. In the case of our study, many people, despite having a higher education, did not follow the recommendations regarding stimulant consumption and physical activity.

An important issue that affects mental well-being is the fact of being diagnosed with the disease, followed by the duration of the disease, the number of diagnosed disease entities, and the acceptance of chronic disease. These dependencies directly affect the patient—the longer the duration and the more diagnosed diseases, the worse the quality-of-life parameters assessed by people treated for cardiovascular diseases [38,39]. In the present study, patients had one to four different cardiovascular diseases, that is, atherosclerosis, hypertension, ischemic heart disease, and infarcts of various organs. Statistically, significantly more men had atherosclerosis, while the frequency of the other diseases was similar for both men and women. Accepting and coming to terms with having a chronic disease reduces the intensity of negative emotions associated with this problem and thus relieves stress. A positive attitude among patients also influences a better assessment of the treatment and greater involvement in the treatment process, for example, by following medical recommendations. However, there are a few reports on the study of the level of acceptance of diseases in people with cardiovascular disorders [40]. As a result of our research, we analyzed the emotional states of patients. One group was characterized by a lower willingness to live and a lack of happiness, but without a breakdown related to the disease. The second cluster included people indicating limitations in their functioning. They are concerned with the limitations resulting from the disease, subjective assessment of health, frequency of social contacts, and mental feelings determined by the disease.

### 4.3. Biochemical Parameters

#### Vitamin Concentrations and Lipid Profiles

The increased risk of cardiovascular disease may result from an inadequate supply of antioxidant vitamins. These vitamins help the body remove free radicals that initiate damage at the cellular level and play an essential role in forming oxidatively modified LDL, substances with pro-atherogenic properties that contribute to the formation of atherosclerosis [41,42]. Many studies have shown that a high intake of antioxidant vitamins and a high concentration in blood serum reduce the likelihood of heart disease [43,44].

Observational studies have found inverse associations between vitamins A and E and risk of atherosclerotic cardiovascular disease (ASCVD). On the other hand, the intervention trials have failed to confirm these findings. Also, trials of supplementation with B vitamins (B6, folic acid, and B12), and vitamin C have not shown beneficial effects. There was also no benefit from vitamin D supplementation regarding cardiovascular complications [45,46].

We have shown that more than 50% of patients are deficient in 25-OH-D3 and retinol. Most patients also had significant deficiencies in α-tocopherol and β-carotene. Similar results have been obtained in other studies conducted in patients with cardiovascular disease [16,47,48,49] and healthy subjects [50,51,52].

A deficiency of β-carotene was particularly significant in our study. In the studies of Miranda et al. [47], on the relationship between the severity of ischemic heart disease and plasma vitamin levels, a deficiency of α-tocopherol and retinol was found in some patients. In contrast, lower concentrations of these vitamins were observed in patients at the initial stage of the disease.

Low concentrations of fat-soluble vitamins are also found in people without cardiovascular disease. Mata-Granados et al. [50] observed retinol, α-tocopherol, and 25-OH-D3 deficiency in healthy individuals of the Spanish population. Only 17% of the subjects had normal vitamin D levels. Additionally, it was found that women had significantly lower concentrations of retinol, α-tocopherol, and 25-OH-D3 than men. In addition, there has been a link between vitamin D levels in the body and seasonal mood disorders, depression, and premenstrual syndrome [53]. Postmenopausal women are at risk of deficient vitamin D levels [52]. An association has been found between vitamin D and sex hormones in postmenopausal women, where the reduction of estrogen levels and other hormonal changes causes a tendency to develop low levels of vitamin D [54]. According to Kand’ar et al. [49], concentrations of fat-soluble vitamins are age-dependent. Lower values were observed in older patients [49,55]. A similar relationship was demonstrated by Franzke et al. [56] in the population of people >65 years of age without cardiovascular diseases. β-carotene deficiency was observed in 73%, α-tocopherol in 33%, and 25-OH-D3 in 61% of the patients.

The occurrence of lipid disorders is also closely related to the discussed cardiovascular risk factors. As patients with established CVD belong to a very high cardiovascular risk group, their LDL goal should be <55 mg/dL [46]. We showed this value was above the recommended level in most subjects. In addition, about 40% of patients had abnormal TG and TC values, and more than 20% had HDL cholesterol levels below the reference value. Drożdż et al. [57] obtained similar results by analyzing the lipid profiles of healthy subjects and patients with cardiovascular diseases in the Polish population from rural areas. Abnormal HDL values were found in 30% of patients and 19% of healthy subjects, while in 40% of patients and 27% of healthy subjects, TG levels were ≥150 mg/dL [57]. It should be emphasized that the occurrence of lipid disorders is closely related to risk factors such as age, gender, stimulants, low levels of physical activity, obesity, diabetes, and psychosocial factors [56]. The agglomeration method was used to reduce and classify the data obtained from the interview. Subjects were assigned to two clusters based on the prevalence of cardiovascular disease. Statistically significant differences were found for all the diseases analyzed. The analysis of the plasma concentrations of fat-soluble vitamins, depending on the analyzed disease entity, showed a significantly higher concentration of retinol in patients with atherosclerosis. The level of other analyzed vitamins in this group of people was lower than that of those without atherosclerosis, but there were no significant differences. We observed an analogous situation for the other analyzed disease entities. Data obtained from many meta-analyses indicate that low vitamin D status is associated with a significantly increased risk of hypertension, cardiovascular events, and cardiovascular mortality [58,59,60]. Data indicate a nonlinear increase in CVD risk at 25-OH-D concentrations below 50 nmol/L compared with a 25-OH-D concentration around 75 nmol/L, with the highest CVD risk at concentrations below 25 nmol/L [61]. Several clinical investigations were also focused on the effect of γ-tocopherol, which is inversely correlated with coronary heart disease [62,63].

In addition, no significant differences were found in the lipid profiles of individuals depending on their disease and assignment to a given cluster. However, it should be added that the main component of treatment was statins and/or other drugs affecting lipid components, such as ezetimibe.

### 4.4. Lifestyle Factors

The analysis of selected lifestyle factors (physical activity and the use of stimulants) did not show statistically significant differences between the studied clusters regarding plasma vitamin concentrations and lipid profile parameters. It is known that an increase in physical activity improves parameters important for cardiovascular diseases, such as weight reduction, increased carbohydrate metabolism, and lower average blood pressure values in people suffering from hypertension, and has a positive effect on the lipid profile [64]. The physical activity of patients participating in the study was average, which may be related to age and vitamin and mineral deficiencies—especially vitamin D3 and calcium, which in old age result in bone mass losses, limiting physical function [65].

Alcohol abuse is another factor that increases the risk of cardiovascular diseases, such as stroke and hypertension, and significantly weakens the effect of antihypertensive drugs [66]. Alcohol consumption of 30 g of ethanol per day also increases triglycerides, homocysteine, and HDL cholesterol [67]. We showed that most of the people surveyed had drunk alcohol occasionally or in the past. Almost half of the respondents were addicted to smoking cigarettes, and practically, as many reported having smoked in the past (Table 4). Nicotine use contributes to about 30% of all deaths from cardiovascular disease in Western countries [68]. The risk of death from cigarette smoking increases with the simultaneous occurrence of hypertension, hypercholesterolemia, peripheral vascular disease, glucose intolerance, and diabetes, and the risk of coronary heart disease among passive smokers increases by 20–30% [68,69].

Patients assigned to the analyzed clusters based on their health and functioning status indicated the consequences of the disease regarding daily activities. First of all, fatigue and a worse mood were reported. No significant differences between the clusters were reported depending on the concentration of the vitamins studied. On the other hand, statistically significant differences were found concerning the lipid profile parameters, that is, %HDL, LDL, and non-HDL values.

The presence of negative emotions (sadness, anxiety, fatigue, lack of willingness to act, etc.) among cardiac patients should be a reason to undertake tests aimed at diagnosing or excluding depression. In the case of people with cardiovascular diseases, the occurrence of depression is a factor that increases the risk of myocardial infarction and death, as it contributes to the intensification of atherosclerotic processes [5,70].

### 4.5. Food Preferences

Proper nutrition combined with a healthy lifestyle is a determinant of maintaining health. The link between a well-balanced diet and a reduced risk of cardiovascular disease is well-known [71,72]. Our analysis of preferences in terms of consumption of selected groups of food products showed significant differences between the studied clusters in addition to the consumption of meat and meat products, dairy products, and pasta. In the context of the conducted research, this seems particularly important in the case of such product groups as fish, eggs, groats, legumes, vegetables, and fruits. It also appears that, from a nutritional point of view, patients’ preferences in cluster 2 were more “health-promoting” in nature. Studies conducted by other authors concerning the eating behavior of people with cardiovascular diseases indicate irregular eating behavior, including infrequent consumption of groats, legumes, fish, vegetables, and fruits. The consequence of this is insufficient intake of dietary fiber and essential vitamins and minerals, deficiencies of which increase the risk of cardiovascular disease [72,73]. Taking into account the fact that eating one serving of vegetables (77 g) reduces the risk of developing cardiovascular diseases by 4% and consuming a portion of fruit weighing 80 g by 5%, it is reasonable for people in the group of patients with these diseases (and not only) to enrich their diet with these groups of food products [43].

The data analysis on plasma vitamin concentrations in the clusters we studied showed significant differences only in the case of β-carotene. Clinically alarming, plasma concentrations of retinol, α-tocopherol, and β-carotene were below the reference values. Therefore, increasing the consumption of food products containing adequate amounts of vitamins, including vitamins that have a protective effect against the risk of developing cardiovascular diseases, is particularly important. For this reason, it seems reasonable to increase the awareness of the studied group of patients regarding higher consumption of vegetables and fruits and thus a change in eating habits [74].

A statistically significant difference was found for β-carotene and α-tocopherol concentrations. Significantly higher levels of β-carotene and α-tocopherol occurred in the group in cluster 2, characterized by a higher consumption of fruit and vegetables.

Food preferences did not significantly affect total cholesterol, HDL and LDL cholesterol, and triglyceride levels. The study’s results do not correspond with the results of other authors, who indicate that the lipid profile parameters of patients with cardiovascular diseases are related to their dietary preferences. Consumption of meat, animal fats, and industrially hardened oils affects the deterioration of lipid profile parameters and thus increases the risk of cardiovascular diseases [75].

### 4.6. Limitations

Our study had several limitations. The most important of these was the small size of the study group, the lack of a control group, and the disproportionate number of women relative to men, which probably contributed to the failure to reveal many other relationships between the parameters studied. Additionally, the study did not take into account patients’ use of supplementation, duration of sun exposure during the day, or use of SPF filters. A strength of the study was the use of multiple predictors to determine psychosocial and dietary behaviors and the analysis of concentrations of selected vitamins in patients with cardiovascular disease.

## 5. Conclusions

No significant relationship was found between the levels of α-tocopherol, β-carotene, and 25-OH-D3 in plasma and lipid profile parameters and cardiovascular diseases, that is, atherosclerosis, hypertension, ischemic heart disease, and myocardial infarction. In contrast, significantly higher retinol levels were found in those with atherosclerosis. The observed reduced levels of α-tocopherol and β-carotene in the study groups and statistically significant differences in retinol levels in patients with atherosclerosis compared to those without the condition were probably not clinically significant due to the small group size. The influence of psychosocial factors and changes in quality of life resulting from cardiovascular disease was demonstrated. The disease duration, the number of diagnosed disease entities, and the impediments to functioning affected psychological well-being. The negative impact of the disease on selected quality-of-life parameters was noted.

Cluster analysis showed variation in preferences in selected clusters, in which patients were characterized by more or less health-promoting eating behaviors considering the consumption of selected food groups. The observed differences were clinically referenced in the plasma concentrations of the analyzed vitamins, β-carotene, and tocopherol, and the lipid profile regarding % HDL. The other analyzed biochemical parameters did not reflect the observed abnormalities in dietary behavior.

It seems reasonable to intensify educational activities to increase cardiac patients’ knowledge. To increase the effectiveness of preventing and treating cardiovascular diseases, the need for interdisciplinary cooperation between doctors, psychologists, and specialists in human nutrition seems reasonable. Given the small patient count, further research is needed.

## Figures and Tables

**Figure 1 nutrients-16-01866-f001:**
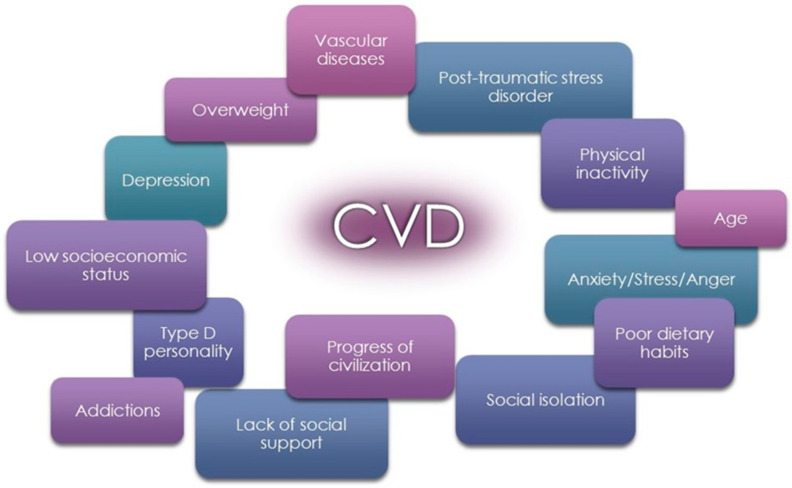
Psychosocial risk factors for cardiovascular diseases (CVD).

**Figure 2 nutrients-16-01866-f002:**
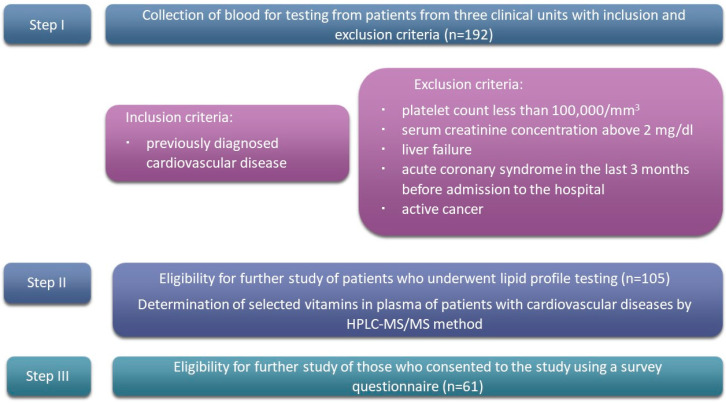
Research scheme.

**Figure 3 nutrients-16-01866-f003:**
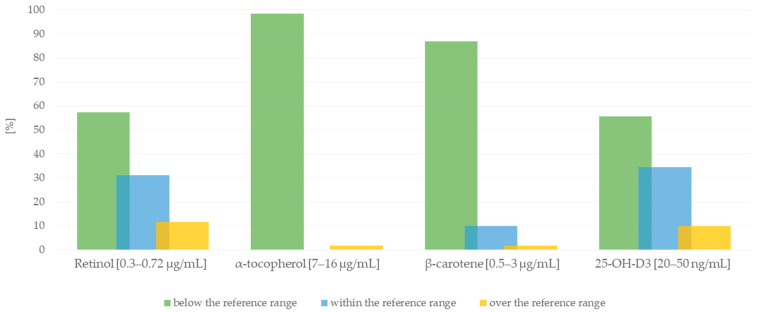
Percentage distribution of retinol, α-tocopherol, β-carotene, and 25-OH-D3 plasma concentrations in the study group of patients in relation to reference values.

**Figure 4 nutrients-16-01866-f004:**
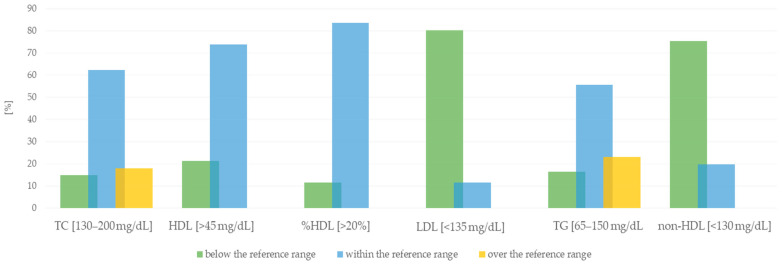
Percentage distribution of lipid profile parameter values in relation to reference values in the study group.

**Table 1 nutrients-16-01866-t001:** Baseline characteristics of the study population.

			Female $\bar{x}$ ± SD		Female $\bar{x}$ ± SD		*p*-Value
			*n*	%	*n*	%	
			22	36	39	64	**0.040 ^a^**
Parameter analyzed	Age (years)		65.0 ± 9.13		61.1 ± 3.17		>0.05 ^b^
	Weight (kg)		83.5 ± 13.6		83.0 ± 10.0		>0.05 ^b^
	Height (m)		1.70 ± 0.085		1.73 ± 0.094		>0.05 ^b^
	BMI (kg/m^2^)		28.9 ± 3.23		27.6 ± 1.89		**0.036 ^b^**
Place of residence	Village		6	27	9	23	**0.045 ^b^**
	City		16	73	30	77	
Parameter analyzed	Primary		3	5	6	10	>0.05 ^b^
	Vocational		6	10	11	18	
	Secondary		3	5	5	8	
	Higher		10	16	17	28	
Number of disease units diagnosed	1		8	36	13	33	>0.05 ^b^
	2		11	50	18	46	
	3		3	14	7	18	
	4		0	0	1	3	
Diagnosis	Atherosclerosis	No	22	100	31	79	**0.042 ^c^**
		Yes	0	0	8	21	
	Arterial hypertension	No	4	18	12	31	>0.05 ^c^
		Yes	18	82	27	69	
	Ischemic heart disease	No	5	23	11	28	>0.05 ^c^
		Yes	17	77	28	72	
	Infarctions of various organs	No	18	82	28	72	>0.05 ^c^
		Yes	4	18	11	28	
Duration of illness	2–3 years		4	18	16	41	>0.05 ^d^
	4–5 years		9	41	12	31	
	Over 5 years		9	41	11	28	

Legend: N—number of participants; $\bar{x}$ ± SD—mean values ± standard deviation; ^a^ χ^2^ test for observed versus expected counts; ^b^ Student’s *t*-test (calculated from raw data); ^c^ χ^2^ test (with Cochran–Mantel–Haenszel correction); ^d^ Pearson’s χ^2^ test; Bold values denote statistical significance at the *p* < 0.05 level.

**Table 2 nutrients-16-01866-t002:** The results of plasma vitamin concentrations and analysis of lipid profiles in groups of patients were divided according to sex.

Parameter Analyzed	Sex	N	M ± SD Me ± IQR ^#^	*p*-Value
25-OH-D3 [20–50 ng/mL] *	Male	39	22.3 ± 15.2	>0.05 ^b^
	Female	22	17.5 ± 17.6	
25-OH-D2 [µg /mL] ** ^#^	Male	39	5.80 ± 3.39	>0.05 ^b^
	Female	22	5.00 ± 1.56	
Retinol [0.3–0.72 µg/mL] *	Male	39	0.43 ± 0.51	**0.021 ^b^**
	Female	22	0.30 ± 0.32	
α-tocopherol [7–16 µg/mL] *	Male	39	2.65 ± 4.65	>0.05 ^b^
	Female	22	0.50 ± 1.19	
β-carotene [0.5–3 µg/mL] *	Male	39	0.30 ± 0.87	>0.05 ^b^
	Female	22	0.08 ± 0.35	
TC [130–200 mg/dL] ^#^	Male	39	175 ± 49.4	>0.05 ^a^
	Female	22	164 ± 32.8	
HDL [>45 mg/dL] *	Male	39	52.2 ± 13.4	**0.027 ^a^**
	Female	22	61.2 ± 16.7	
%HDL [>20%] *	Male	39	31.6 ± 10.0	**0.016 ^a^**
	Female	22	38.2 ± 9.90	
LDL [<135 mg/dL] *	Male	39	95.5 ± 41.1	>0.05 ^b^
	Female	22	79.8 ± 28.9	
TG [65–150 mg/dL] * ^#^	Male	39	141 ± 112.2	>0.05 ^b^
	Female	22	103 ± 68.6	
Non-HDL [<120 mg/dL] * ^#^	Male	39	123 ± 49.3	>0.05 ^b^
	Female	22	115 ± 31.1	

Legend: N—number of patients; M ± SD—mean values ± standard deviation; ^#^—Me ± IQR—median ± interquartile range; ^a^ Student’s *t*-test; ^b^ Mann–Whitney U-test—two-sided; bold values denote statistical significance at the *p* < 0.05 level. *—reference ranges; **—no reference ranges.

**Table 3 nutrients-16-01866-t003:** The results of plasma vitamin concentrations and analysis of lipid profiles in groups of patients were divided according to the criterion of the presence of atherosclerotic lesions.

Parameter Analyzed	Atherosclerosis	N	M ± SD Me ± IQR ^#^	*p*-Value
25-OH-D3 [20–50 ng/mL] *	No	53	23.2 ± 16.6	>0.05 ^a^
	Yes	8	19.3 ± 10.9	
25-OH-D2 [µg /mL] ** ^#^	No	53	5.70 ± 2.96	>0.05 ^b^
	Yes	8	5.78 ± 2.21	
Retinol [0.3–0.72 µg/mL] *	No	53	0.31 ± 0.44	**0.016 ^a^**
	Yes	8	0.59 ± 0.49	
α-tocopherol [7–16 µg/mL] *	No	53	2.14 ± 4.08	>0.05 ^a^
	Yes	8	1.90 ± 1.56	
β-carotene [0.5–3 µg/mL] *	No	52	0.28 ± 0.78	>0.05 ^a^
	Yes	8	0.15 ± 0.21	
TC [130–200 mg/dL] ^#^	No	52	165 ± 33.0	>0.05 ^b^
	Yes	7	159 ± 41.3	
HDL [>45 mg/dL] *	No	52	57.0 ± 15.0	>0.05 ^a^
	Yes	7	45.0 ± 8.00	
%HDL [>20%] *	No	52	34.5± 10.5	>0.05 ^a^
	Yes	7	30.9 ± 10.4	
LDL [<135 mg/dL] *	No	49	89.1 ± 38.3	>0.05 ^b^
	Yes	7	85.0 ± 36.1	
TG [65–150 mg/dL] * ^#^	No	52	100 ± 74.2	>0.05 ^b^
	Yes	7	148 ± 65.0	
Non-HDL [<120 mg/dL] * ^#^	No	52	100 ± 36.0	>0.05 ^b^
	Yes	7	121 ± 40.5	

Legend: N—number of patients; M ± SD—mean values ± standard deviation; ^#^—Me ± IQR—median ± interquartile range; ^a^ Student’s *t*-test; ^b^ Mann–Whitney U-test—two-sided; bold values denote statistical significance at the *p* < 0.05 level. *—reference ranges; **—no reference ranges.

**Table 4 nutrients-16-01866-t004:** Analysis of plasma concentrations of selected vitamins and lipid profiles in groups of patients divided according to the criteria of selected lifestyle factors.

Parameter Analyzed	Cluster	N	M ± SD Me ± IQR ^#^	*p*-Value
25-OH-D3 [20–50 ng/mL] *	1	28	20.8 ± 12.5	>0.05 ^a^
	2	33	24.4 ± 14.2	
25-OH-D2 [µg /mL] ** ^#^	1	28	5.00 ± 1.04	>0.05 ^b^
	2	33	5.55 ± 2.71	
Retinol [0.3–0.72 µg/mL] *	1	28	0.37 ± 0.25	>0.05 ^a^
	2	33	0.34 ± 0.24	
α-tocopherol [7–16 µg/mL] *	1	28	1.62 ±0.78	>0.05 ^a^
	2	33	2.53 ± 1.04	
β-carotene [0.5–3 µg/mL] *	1	28	0.16 ± 0.11	>0.05 ^a^
	2	32	0.31 ± 0.23	
TC [130–200 mg/dL] ^#^	1	27	162 ± 41.0	>0.05 ^a^
	2	32	164 ± 39.2	
HDL [>45 mg/dL] *	1	27	58.0 ± 17.0	>0.05 ^a^
	2	32	53.0 ± 8.00	
%HDL [>20%] *	1	27	34.8 ± 10.7	>0.05 ^a^
	2	32	33.5 ± 10.3	
LDL [<135 mg/dL] *	1	25	88.1 ± 39.5	>0.05 ^a^
	2	31	89.0 ± 36.9	
TG [65–150 mg/dL] * ^#^	1	27	103 ± 86.4	>0.05 ^a^
	2	32	100 ± 88.0	
Non-HDL [<120 mg/dL] * ^#^	1	27	162 ± 40.0	>0.05 ^a^
	2	32	164 ± 43.5	

Legend: N—number of patients; M ± SD—mean values ± standard deviation; ^#^—Me ± IQR—median ± interquartile range; ^a^ Student’s *t*-test; ^b^ Mann–Whitney U-test—with correction for continuity; cluster 1—as-signed non-smokers and past smokers; cluster 2—occasional smokers and addicts; *—reference ranges; **—no reference ranges.

**Table 5 nutrients-16-01866-t005:** Analysis of vitamin concentrations and lipid profiles in groups of patients divided according to the criterion of indicated current health status in each cluster after applying data reduction.

Parameter Analyze	Cluster	N	M ± SD Me ± IQR ^#^	*p*-Value
25-OH-D3 [20–50 ng/mL] *	1	17	19.0 ± 13.3	>0.05 ^a^
	2	44	24.2 ± 16.8	
25-OH-D2 [µg /mL] ** ^#^	1	17	5.06 ± 1.76	>0.05 ^a^
	2	44	5.18 ± 2.01	
Retinol [0.3–0.72 µg/mL] *	1	17	0.48 ± 0.39	>0.05 ^a^
	2	44	0.31 ± 0.27	
α-tocopherol [7–16 µg/mL] *	1	17	1.52 ±1.40	>0.05 ^a^
	2	44	2.34 ± 1.89	
β-carotene [0.5–3 µg/mL] *	1	17	0.17 ± 0.15	>0.05 ^a^
	2	43	0.33 ± 0.29	
TC [130–200 mg/dL] ^#^	1	17	183 ± 41.5	>0.05 ^b^
	2	42	161 ± 39.6	
HDL [>45 mg/dL] *	1	17	53.0 ± 12.0	>0.05 ^b^
	2	42	57.0 ± 16.0	
%HDL [>20%] *	1	17	29.5 ± 9.56	**0.032 ^a^**
	2	42	35.9 ± 10.3	
LDL [<135 mg/dL] *	1	16	105 ± 41.6	**0.044 ^b^**
	2	40	82.0 ± 34.5	
TG [65–150 mg/dL] * ^#^	1	17	138 ± 98,4	>0.05 ^b^
	2	42	94.0 ± 77.0	
Non-HDL [<120 mg/dL] * ^#^	1	17	131 ± 41.3	**0.019 ^b^**
	2	42	97.0 ± 41.7	

Legend: N—number of patients; M ± SD—mean values ± standard deviation; ^#^—Me ± IQR—median ± interquartile range; ^a^ Mann–Whitney U-test—two-sided; ^b^ Student’s *t*-test; bold values denote statistical significance at the *p* < 0.05 level; cluster 1—lower concentrations of 25-OH-D3, α-tocopherol, β-carotene, higher concentrations of retinol, higher %HDL, and lower LDL and non-HDL values; cluster 2—higher concentrations of 25-OH-D3, α-tocopherol, β-carotene, lower concentrations of retinol, lower %HDL, and higher LDL and non-HDL values; *—reference ranges; **—no reference ranges.

**Table 6 nutrients-16-01866-t006:** Analysis of vitamin concentrations in groups of patients and lipid profile divided according to the criterion of indicated current health status in each cluster after applying data reduction.

Parameter Analyzed	Cluster	N	M ± SD Me ± IQR ^#^	*p*-Value
25-OH-D3 [20–50 ng/mL] *	1	26	24.2 ± 18.2	>0.05 ^a^
	2	35	21.7 ± 14.3	
25-OH-D2 [µg /mL] ** ^#^	1	26	5.00 ± 2.11	>0.05 ^a^
	2	35	5.00 ± 3.25	
Retinol [0.3–0.72 µg/mL] *	1	26	0.28 ± 0.35	>0.05 ^a^
	2	35	0.40 ± 0.53	
α-tocopherol [7–16 µg/mL] *	1	26	2.76 ±1.57	>0.05 ^a^
	2	35	1.63 ± 1.52	
β-carotene [0.5–3 µg/mL] *	1	26	0.19 ± 0.11	>0.05 ^a^
	2	34	0.30 ± 0.22	
TC [130–200 mg/dL] ^#^	1	26	170 ± 41.6	>0.05 ^b^
	2	33	156 ± 34.0	
HDL [>45 mg/dL] *	1	26	55.0 ± 15.0	>0.05 ^b^
	2	33	56.0 ± 15.0	
%HDL [>20%] *	1	26	32.6 ± 11.0	>0.05 ^b^
	2	33	35.3 ± 9.92	
LDL [<135 mg/dL] *	1	25	93.6 ± 36.4	>0.05 ^a^
	2	31	84.6 ± 39.0	
TG [65–150 mg/dL] * ^#^	1	26	103 ± 92.1	>0.05 ^a^
	2	33	98.0 ± 86.4	
Non-HDL [<120 mg/dL] * ^#^	1	26	170 ± 41.3	>0.05 ^b^
	2	33	156 ± 42.8	

Legend: N—number of patients; M ± SD—mean values ± standard deviation; ^#^—Me ± IQR—median ± interquartile range; ^a^ Student’s *t*-test; ^b^ Mann–Whitney *U* test—two-sided; cluster 1—no significant impact of health status on functioning; cluster 2—fast fatigue, worse mood, problems with performing daily activities, need to reduce working time; *—reference ranges, **—no reference ranges.

**Table 7 nutrients-16-01866-t007:** Analysis of comparison of consumption preferences of selected food groups across clusters using data reduction *.

Foods	Preferences *	Cluster 1		Cluster 2		*p*-Value ^a^
		N	% N	N	% N	
		26	42.6	35	57.4	
Meat and meat products	I don’t like	1	1.64	0	0.00	>0.05
	I don’t care	0	0.00	1	1.64	
	I like	15	24.6	14	23.0	
	I like it a lot	10	16.4	20	32.8	
Milk and dairy products	I don’t like	0	0.00	2	3.3	>0.05
	I don’t care	1	1.64	1	1.64	
	I like	14	23.0	9	14.8	
	I like it a lot	11	18.0	23	37.7	
Fish	I don’t like	3	4.92	1	1.64	**<0.001**
	I don’t care	6	9.84	2	3.28	
	I like	15	24.6	17	27.9	
	I like it a lot	2	3.28	15	24.6	
Eggs	I don’t like	0	0.00	0	0.00	**0.025**
	I don’t care	2	3.28	2	3.28	
	I like	18	29.5	14	23.0	
	I like it a lot	6	9.84	19	31.2	
White Bread	I don’t like	1	1.64	6	9.84	**<0.001**
	I don’t care	1	1.64	3	4.92	
	I like	22	36.1	17	27.9	
	I like it a lot	2	3.28	9	14.8	
Brown bread	I don’t like	1	1.64	0	0.00	**<0.05**
	I don’t care	13	21.3	3	4.92	
	I like	10	16.4	16	26.2	
	I like it a lot	19	31.2	16	26.2	
Noodles	I don’t like	1	1.64	2	3.28	>0.05
	I don’t care	12	19.7	10	16.4	
	I like	13	21.3	19	31.2	
	I like it a lot	0	0.00	4	6.56	
Groats	I don’t like	9	14.8	1	1.64	**<0.001**
	I don’t care	16	26.2	2	3.28	
	I like	1	1.64	25	41.0	
	I like it a lot	0	0.00	7	11.5	
Legumes	I don’t like	8	13.1	1	1.64	**0.003**
	I don’t care	16	26.2	8	13.1	
	I like	2	3.28	23	37.7	
	I like it a lot	0	0.00	3	4.92	
Vegetables	I don’t like	0	0.00	0	0.00	**<0.001**
	I don’t care	1	1.64	1	1.64	
	I like	22	36.1	16	26.2	
	I like it a lot	3	4.92	18	29.5	
Fruits	I don’t like	0	0.00	2	3.28	**0.001**
	I don’t care	2	3.28	1	1.64	
	I like	21	34.4	14	23.0	
	I like it a lot	3	4.92	18	29.5	

Legend: N—patient count; ^a^ Pearson’s χ^2^ test; bold values denote statistical significance at the *p* < 0.05 level; cluster 1—preferred (liked and very much liked) meat and meat products, milk and dairy products, and light and dark bread, and dislikers and indifferent were for groats, legumes, and dark bread; cluster 2—patients who preferred meat and meat products, milk and dairy products, and higher consumption of fish, groats, legumes, vegetables, and fruits. * The table omits the “I very dislike/I don’t know” preference due to a lack of responses.

**Table 8 nutrients-16-01866-t008:** Analysis of plasma concentrations of selected vitamins and lipid profiles in groups of patients was divided according to the criterion of dietary preferences in assigned clusters after applying data reduction.

Parameter Analyzed	Cluster	N	M ± SD Me ± IQR ^#^	*p*-Value
25-OH-D3 [20–50 ng/mL] *	1	26	23.2 ± 18.8	>0.05 ^a^
	2	35	22.4 ± 13.8	
25-OH-D2 [µg /mL] ** ^#^	1	26	5.00 ± 1.34	>0.05 ^a^
	2	35	5.00 ± 1.07	
Retinol [0.3–0.72 µg/mL] *	1	26	0.24 ± 0.36	>0.05 ^a^
	2	35	0.43 ± 0.39	
α-tocopherol [7–16 µg/mL] *	1	26	1.47 ±1.32	>0.05 ^a^
	2	35	2.59 ± 1.54	
β-carotene [0.5–3 µg/mL] *	1	26	0.24 ± 0.14	**0.034 ^a^**
	2	34	0.28 ± 0.21	
TC [130–200 mg/dL] ^#^	1	26	161 ± 41.0	>0.05 ^b^
	2	33	168 ± 47.2	
HDL [>45 mg/dL] *	1	26	60.0 ± 16.0	>0.05 ^b^
	2	33	52.0 ± 14.0	
%HDL [>20%] *	1	26	37.6 ± 10.6	**0.041 ^b^**
	2	33	31.3 ± 9.60	
LDL [<135 mg/dL] *	1	25	83.0 ± 31.5	>0.05 ^c^
	2	31	93.1 ± 42.1	
TG [65–150 mg/dL] * ^#^	1	26	86.0 ± 41.0	>0.05 ^c^
	2	33	117 ± 73.9	
Non-HDL [<120 mg/dL] * ^#^	1	26	102 ± 34.1	>0.05 ^c^
	2	33	105 ± 32.6	

Legend: N—number of patients; M ± SD—mean values ± standard deviation; ^#^—Me ± IQR—median ± interquartile range; ^a^ Student’s *t*-test; ^b^ Mann–Whitney *U* test—two-sided; ^c^ Mann–Whitney *U* test; bold values denote statistical significance at the *p* < 0.05 level; cluster 1—higher β-carotene level, %HDL; cluster 2—lower β-carotene level, %HDL; *—reference ranges; **—no reference range.

## Data Availability

The data presented in this study are available on request from the corresponding author due to ethical reasons.

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
