# Peer review of "Selected Psychosocial Factors, Nutritional Behavior, and the Analysis of Concentrations of Selected Vitamins in Patients with Cardiovascular Diseases"

_nutrients, 2024, doi:10.3390/nu16121866_

Round 1

Reviewer 1 Report

Comments and Suggestions for Authors

The authors of this study aimed to: In the group of 61 patients with CVD (atherosclerosis, hypertension, ischemic heart disease, myocardial infarction), to evaluate the associations between psychological and dietary factors and their potential influence on the progression of the disease.

At the end of the introductory chapter, include a paragraph on the purpose of this study.

In the first table, the place of reisedecne, education and number of disease units diagnoses should also be compared using the khi squared test, not the student test. 

Was there a sex difference in vitamin levels? 

How were vitamin deficiencies associated with cardiovascular disease, and were there gender differences? 

Was there a sex difference in lipid profiles? 

How were lipid profiles associated with cardiovascular disease, and were there gender differences? 

Do not mention table, figure numbers and p values in the discourse, that can only be in the results section. 

The results obtained are incomplete, e.g. has it not been tested whether negative emotional state is associated with more severe cardiovascular disease? 

For vitamin deficiencies, the study did not investigate whether different vitamin deficiencies were associated with CVC diseases, or whether there was a sex difference. 

How were the different clusters selected? 

Overall, the manuscript deals with an interesting topic. But it does not reveal to the reader the extent to which different vitamin deficiencies, lipid profiles, psychosocial factors increase cardiovascular factors. There should be a control group to see what proportion of people who do not have cardiovascular disease have these factors. I suggest a major revision to take these into account. 

Author Response

Response to Reviewer 1 Comments:

We appreciate the Reviewer for taking the time to carefully review the manuscript and give constructive and detailed comments, which has greatly helped to improve this paper. Below is our point-by-point response to the comments:

Q1. At the end of the introductory chapter, include a paragraph on the purpose of this study.

A1. Following the Reviewer’s  suggestion we added the purpose of the study (lines 97-100):

The aim of the study was to assess the relationship between psychological and dietary factors in the group of patients with CVD (atherosclerosis, hypertension, ischemic heart disease, myocardial infarction) and their possible impact on the course of the disease.

Q2. In the first table, the place of reisedecne, education and number of disease units diagnoses should also be compared using the khi squared test, not the student test. 

A2. Thank you for this point. Indeed, in the Table 1, nominal data were compared using the chi-square test. We added a misplaced designation. In one place, the p-value was also corrected.

Q3. Was there a sex difference in vitamin levels?

A3.   We compared the data of plasma vitamin concentrations and analysis of lipid profiles in groups divided according to sex. Differences were observed between men and women in retinol, HDL, and %HDL levels (Table 2). Comment was added in the lines 230-232.

Q4. How were vitamin deficiencies associated with cardiovascular disease, and were there gender differences?

A4. As presented in Table 4, we observed differences between patients divided according to arteriosclerosis lesions, but only at the retinol level. The plasma vitamin concentrations and lipid profiles did not differ between patients divided according to the criterion of the presence of arterial hypertension, ischemic heart disease, and infarcts of various organs. We added these comments to the result section (Lines 268-271).

Q5. Was there a sex difference in lipid profiles?

A5. As mentioned above, differences were observed between men and women in retinol, HDL, and %HDL levels (Table 2). Comment was added in the lines 230-230.            

Q6. How were lipid profiles associated with cardiovascular disease, and were there gender differences? 

A6. Taking together the conjunction of sex and atherosclerosis, arterial hypertension, ischemic heart disease, and infarcts of various organs, we observed differences only in atherosclerosis and Ischemic heart disease in retinol level. In the first case, females without atherosclerosis had lower retinol levels. In the case of ischemic heart disease, men without Ischemic heart disease had significantly higher retinol levels than women with and without the disease. However, we decided not to show this data because of the small number of observations in these groups. We added these comments to the result section (Lines 288-294).

Q7.  Do not mention table, figure numbers and p values in the discourse, that can only be in the results section. 

Q7: Thank you for your comment. Mentioning of table, figure numbers and p values were deleted in the discourse

Q8. The results obtained are incomplete, e.g. has it not been tested whether negative emotional state is associated with more severe cardiovascular disease? 

A8. Thank you for this point. During our analysis, we analyzed separately and in clusters whether negative emotional state is associated with more severe CVD. However, the results was not shown because the selected disorders did not influence the emotional state. The comment was added in the lines 361-363.

      Moreover, it has been tested whether negative emotional state is associated with more severe CVD. The selected disorders did not influence the emotional state as analyzed separately or in clusters (data not shown).

Q9. For vitamin deficiencies, the study did not investigate whether different vitamin deficiencies were associated with CVC diseases, or whether there was a sex difference. 

A9. As presented in Table 4, we observed differences between patients divided according to arteriosclerosis lesions, but only at the retinol level. The plasma vitamin concentrations and lipid profiles did not differ between patients divided according to the criterion of the presence of arterial hypertension, ischemic heart disease, and infarcts of various organs. We added these comments to the result section (Lines 268-270).

Q10. How were the different clusters selected? 

A10. Dear Reviewer, the computer algorithm was used to make the clusters, but we carefully chose the data to cluster it. Ward's method of agglomerating variables and determining clusters based on the similarity of responses with the Euclidean distance determination was used to reduce and classify questionnaire-based patient data. A tree and scree plots were used to determine the number of clusters for each agglomeration. Patients were assigned to individual clusters based on tying common cases into groups. Due to a large amount of data, we analyzed the reduced data using agglomerations of objects and features. Agglomerations forming two to three clusters were used, thus allowing observation of the influence of multiple parameters on each other. Differences in group sizes in the various analyses are due to error and/or non-response.

Q11. Overall, the manuscript deals with an interesting topic. But it does not reveal to the reader the extent to which different vitamin deficiencies, lipid profiles, psychosocial factors increase cardiovascular factors. There should be a control group to see what proportion of people who do not have cardiovascular disease have these factors. I suggest a major revision to take these into account.

A11. Thank you for this point. We agree with the reviewer that the study on the association of vitamin deficiencies, lipid profile and psychosocial factors along with cardiovascular factors and their influence  on the development and progression of CVD would be of interest to the reader. However, it was not the aim of our study. We are aware that we should include a control group to compare what proportion of people who do not have cardiovascular disease have these factors. This aspect could be the subject of our future study.

Sincerely Regards,

The authors.

Reviewer 2 Report

Comments and Suggestions for Authors

The manuscript entitled Selected psychosocial factors, nutritional behavior and the analysis of concentrations of selected vitamins in patients with cardiovascular diseases is an original article. The authors assessed the relationships between psychosocial factors (assessed by author-designed questionnaires.) and nutritional factors in patients with cardiovascular diseases and their possible impact on the course of the disease. They analyzed plasma concentrations of vitamins A, E, D, and β-carotene and the lipid profile enzymatically. The conclusion was that it is important to increase the effectiveness of the prevention and treatment of cardiovascular diseases by interdisciplinary cooperation between doctors, psychologists, and specialists in human nutrition.

The messages of this study are important in clinical practice. This study is very interesting because this theme is still in debate. Even there are data, which show an association between cardiovascular diseases risk, and low dose of these vitamins, there was not proved a direct relationship. In addition, the correlation with psychosocial status and the lipid profile is a great point for this study.

However, I have some comments.

Abdominal circumference is associated with cardiovascular disease risk. Could you include this parameter in the study?

Discussions are difficult to read. I recommend dividing in sub chapters for making it more structured.

Author Response

Response to Reviewer 2 Comments:

We express our gratitude to the Reviewer for dedicating their time to review the manuscript and provide detailed comments, which have contributed to improving this paper.

Q1. Abdominal circumference is associated with cardiovascular disease risk. Could you include this parameter in the study?

A1. Unfortunately, abdominal circumference measurements were not taken.

Q2. Discussions are difficult to read. I recommend dividing in sub chapters for making it more structured.

A2. Thank you for the suggestion. We divided Discussion into the following sections:

      4.1. Anthropometric Parameters

      4.2. Socioeconomic status and Psychosocial Factors

      4.3. Biochemical Parameters

                     4.3.1. Vitamin Concentrations and Lipid Profiles

      4.4. Lifestyle Factors

      4.5. Food preferences

Sincerely Regards,

The authors.
